# Circadian regulation of vertebrate cone photoreceptor function

Jingjing Zang[1], Matthias Gesemann[1], Jennifer Keim[1], Marijana Samardzija[2], Christian Grimm[2], Stephan CF Neuhauss[1]*

[1]University of Zurich, Department of Molecular Life Sciences, Zurich, Switzerland; [2]Lab for Retinal Cell Biology, Department of Ophthalmology, University Hospital Zurich, University of Zurich, Zurich, Switzerland

**Abstract** Eukaryotes generally display a circadian rhythm as an adaption to the reoccurring day/night cycle. This is particularly true for visual physiology that is directly affected by changing light conditions. Here we investigate the influence of the circadian rhythm on the expression and function of visual transduction cascade regulators in diurnal zebrafish and nocturnal mice. We focused on regulators of shut-off kinetics such as Recoverins, Arrestins, Opsin kinases, and Regulator of G-protein signaling that have direct effects on temporal vision. Transcript as well as protein levels of most analyzed genes show a robust circadian rhythm-dependent regulation, which correlates with changes in photoresponse kinetics. Electroretinography demonstrates that photoresponse recovery in zebrafish is delayed in the evening and accelerated in the morning. Functional rhythmicity persists in continuous darkness, and it is reversed by an inverted light cycle and disrupted by constant light. This is in line with our finding that orthologous gene transcripts from diurnal zebrafish and nocturnal mice are often expressed in an anti-phasic daily rhythm.

*For correspondence:
stephan.neuhauss@mls.uzh.ch

**Competing interest:** The authors declare that no competing interests exist.

## Introduction

Circadian rhythms serve as endogenous clocks that molecularly support the daily occurring oscillations of physiology and ensuing behavior (*Brown et al., 2019*; *Cahill, 2002*; *Frøland Steindal and Whitmore, 2019*; *Golombek et al., 2014*; *Idda et al., 2012*; *Ukai and Ueda, 2010*; *Vatine et al., 2011*). It has long been recognized that the central pacemaker of circadian rhythms resides in dedicated brain regions, either the suprachiasmatic nucleus in mammals or the pineal gland in non-mammalian vertebrates. The rhythm is entrained by external stimuli (eg, light) that directly act on the core circadian transcriptional feedback loop. Multiple studies have shown that autonomous circadian clocks also exist in other brain regions and in peripheral tissues (*Frøland Steindal and Whitmore, 2019*; *Idda et al., 2012*; *Vatine et al., 2011*). This is particularly true for the retina, which generates its own circadian rhythm (*Gladys, 2020*). In zebrafish, this rhythmicity is reflected in a number of circadian adaptations, such as a higher response threshold in the morning (*Li and Dowling, 1998*), photoreceptor retinomotor movement in constant darkness (*Menger et al., 2005*), and cone photoreceptor synaptic ribbon disassembly at night (*Emran et al., 2010*). Such adaptations are also found in other animals such as mice, where stronger electrical retinal coupling during the night (*Jin et al., 2015*; *Li et al., 2009*; *Ribelayga et al., 2008*), as well as slower dark adaptation of rods during the day, was observed (*Xue et al., 2015*). The molecular mechanisms underlying these circadian-dependent retinal regulations are still largely unknown.

In the vertebrate retina, there are two different types of photoreceptors, namely rods and cones (*Burns and Baylor, 2001*; *Fu and Yau, 2007*). Rods function mainly during dim light conditions, whereas cones are characterized by lower sensitivity but faster response kinetics, being important for daylight and color vision. About 92 % of larval and 60 % of adult photoreceptors in the zebrafish retina are cones (*Allison et al., 2010*; *Fadool, 2003*; *Zimmermann et al., 2018*). Although rods and cones

generally use the same visual transduction cascade components, the individual reactions are typically mediated by photoreceptor type-specific proteins.

Visual transduction commences by an opsin chromophore-mediated absorption of photons, which triggers the activation of a second messenger cascade including the trimeric G-protein transducin. Activated transducin stimulates the effector enzyme phosphodiesterase (PDE), which leads to a reduction in intracellular cyclic guanosine monophosphate (cGMP) levels, subsequently leading to the closure of cyclic nucleotide -gated (CNG) cation channels, resulting in a membrane potential change (*Fain et al., 2001*; *Lamb and Pugh, 2006*).

High-temporal resolution requires a tightly regulated termination of visual transduction (*Chen et al., 2012*; *Matthews and Sampath, 2010*; *Zang and Matthews, 2012*). This depends on the highly effective quenching of both the activated visual pigment (R*) and the PDE-transducin complex (PDE*). R* is phosphorylated by a G-protein receptor kinase (GRK) before being completely deactivated by binding to arrestin. While GRK activity itself is controlled by recoverin (RCV) in a $Ca^{2+}$-dependent manner (*Zang and Neuhauss, 2018*), the quenching of PDE* depends on the GTPase activity of its γ-subunit that is regulated by activator protein RGS9 (Regulator of G-protein Signaling 9) (*Krispel et al., 2006*).

We now show that the expression levels of these important regulators of cone visual transduction decay are modulated by the circadian clock. Moreover, these periodic fluctuations are reflected in oscillating protein levels that correlate with the rhythmicity in visual physiology and behavior observed in zebrafish. Interestingly, we have found that the expression of a selection of mouse orthologs of the investigated regulatory genes is also modulated by the circadian clock. However, the periodicity was opposite to that of zebrafish, fitting the nocturnal lifestyle of mice.

## Results

### Expression levels of key genes involved in shaping visual transduction decay are regulated by the circadian clock

To determine the influence of the circadian clock on visual behavior, we analyzed gene expression levels of key visual transduction regulators over a 24 hr period using quantitative real-time polymerase chain reacion (qRT-PCR). Eyes from larval (5 days post fertilization [dpf]) and adult zebrafish that were kept under a normal light cycle (LD 14:10, light on at 8 o'clock in the morning), as well as eyes from 5 dpf larvae kept in continuous darkness (DD), were collected every 3 hr over a period of 24 hr and subsequently analyzed. Apart from *rcv2a*, which seems to have no or weak fluctuating transcript levels in larvae (*Figure 1G*), expression levels of the other *recoverins* (*rcv1a*, *rcv1b*, which is absent from larval retina, and *rcv2b*), G-protein receptor kinases (*grk7a* and *grk7b*), arrestins (*arr3a* and *arr3b*), and *regulator of G-protein signaling 9* (*rgs9a*) were clearly oscillating (statistical information in *Supplementary file 1*). In many cases, transcripts were most abundant at ZT1 or ZT4 (*grk7a, grk7b, rcv2b, arr3a*, and *arr3b*), subsequently declined throughout the day, and recovered during the night. For instance, in adult zebrafish eyes, *grk7a* expression levels decreased by around 98 % from the peak to the lowest expression level (*Figure 1A*). In the case of adult *rgs9a*, transcripts reached the highest level at ZT22, with the value very close to ZT1. In situ hybridization (ISH) analysis using digoxigenin-labeled RNA probes validated our qRT-PCR results (*Figure 1—figure supplement 2* and *Figure 1—figure supplement 3*).

Interestingly, two genes, namely *rcv1a* and *rcv2a*, displayed different expression profiles in larval and adult eyes (*Figure 1E&G*). While larval *rcv1a* mRNA transcript levels peaked around ZT19, larval *rcv2a* transcript expression was weak/non-cyclic. However, this is in contrast to adult retinas where *rcv1a* and *rcv2a* transcripts were highest at ZT7 (*Figure 1G*). An anti-phasic expression profile between larval and adult stages can also be observed for rod *arrestins* (*arras*) (*Figure 1—figure supplement 4*).

In order to establish that the daily expression changes of these transcripts are indeed regulated by the intrinsic circadian clock, we repeated our experiments in larvae kept in complete darkness (DD), eliminating light as an external factor. Under normal LD, as well as DD conditions, we obtained largely comparable results (*Figure 1*). Exceptions were *arr3a* and *arr3b*, showing a 3 -hr phase shift, and *rcv1a*, showing an almost anti-phase relationship (see 'Discussion' section).

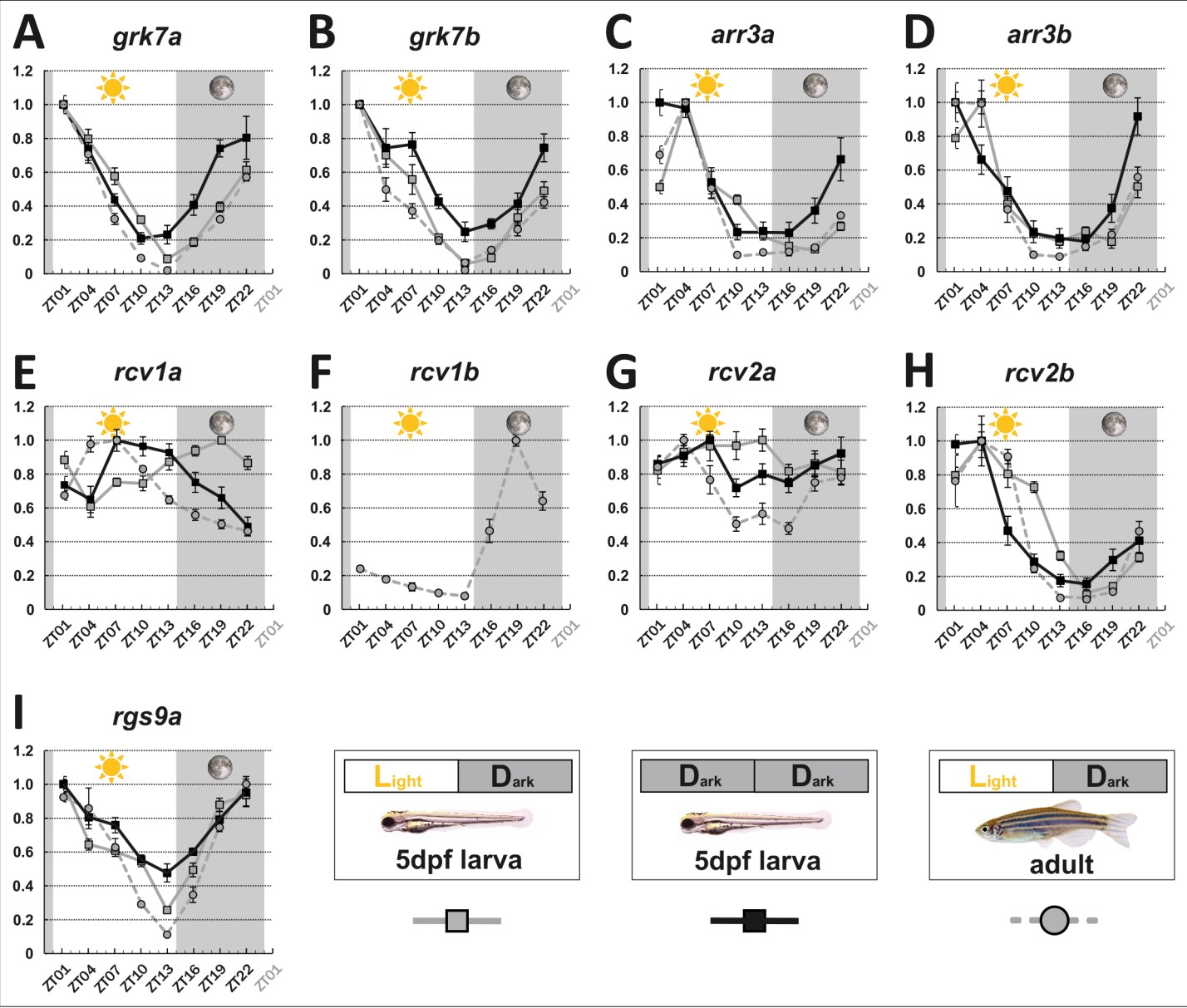

**Figure 1.** Key visual transduction decay gene transcripts that are under circadian control. mRNA levels of visual transduction decay genes in the eye of adult and larval zebrafish were measured by qRT-PCR over a 24-hour-period. (**A-I**). Eye tissues from larval fish either raised under a normal light/dark cycle (LD / gray squares) or in continuous darkness (DD / black squares) and from adult LD zebrafish (gray circles) were collected at eight different time points throughout the day. The name of the analyzed gene transcripts is given on top of each graph. The time point of collection is indicated along the x-axis with ZT01 being the time point one hour after the light was turned on. Dark periods are indicated by the moon symbol and highlighted in gray, whereas the periods under regular light conditions are indicated by the sun symbol and shown in white. For better orientation the different conditions are summarized at the bottom of the figure. Data represents the mean ± standard error of the mean (s.e.m). Statistical analysis was performed by "RAIN" as previously described (**Thaben and Westermark, 2014**). Statistics information and the numbers of independent repeats are provided in **Supplementary file 1**. Metadata can be downloaded from DRYAD.

The online version of this article includes the following figure supplement(s) for figure 1:

**Source data 1.** mRNA levels of visual transduction decay genes in the eye of adult and larval zebrafish were measured by qRT-PCR over a 24 hr period.

**Figure supplement 1.** ISH of *rcv1a* and *rcv2b* as examples indicating the staining in pineal gland, may not or may be synchronized with the staining in the eye.

**Figure supplement 2.** ISH showing different gene expressions at varying time points in 5 dpf zebrafish larvae in dorsal view.

**Figure supplement 3.** ISH showing different gene expressions on radial sections of adult zebrafish retina at different time points indicated on top.

**Figure supplement 4.** ISH showing different rod gene expressions in zebrafish larval and adult retina at different time points indicated on top.

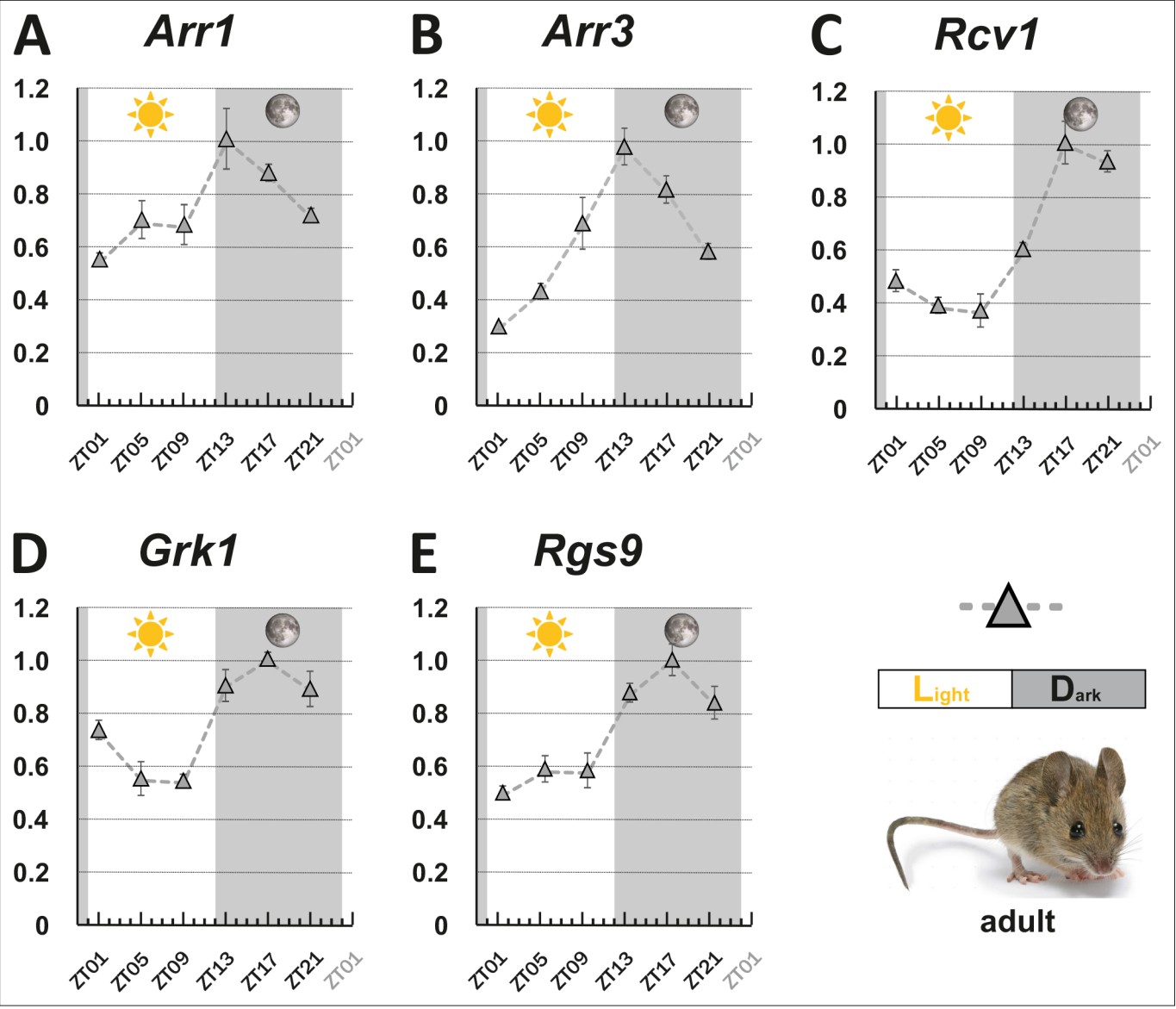

**Figure 2.** Circadian regulation of key visual transduction genes in nocturnal mice is reversed. Transcript levels of indicated mouse genes (**A-E**) were measured using qRT-PCR on retinal tissue of 12-week-old wildtype mice. were measured using qRT-PCR on retinal tissue of 12-week-old wildtype mice. The time point of collection is indicated along the x-axis with ZT01 being the time point one hour after the light was turned on. Dark periods are indicated by the moon symbol and highlighted in gray, whereas the periods under regular light conditions are indicated by the sun symbol and shown in white. Data represents the mean ± s.e.m. Statistical analysis was performed by "RAIN" as previously described (**Thaben and Westermark, 2014**). Statistics information and the numbers of independent repeats are provided in **Supplementary file 2**. Metadata can be downloaded from DRYAD.

The online version of this article includes the following figure supplement(s) for figure 2:

**Source data 1.** mRNA levels of visual transduction decay genes in mouse eyes were measured by qRT-PCR over a 24 hr period.

## Corresponding retinal genes in nocturnal mice display an anti-phasic expression pattern

As zebrafish are diurnal animals having a cone-dominant retina, we wondered if the observed circadian regulation of visual transduction gene transcripts is also seen in the rod-dominant retina of nocturnal mice. We selected mouse *Grk1*, the only visual grk gene in mice (**Chen et al., 1999**; **Wada et al., 2006**), the sole recoverin (**Chen et al., 2012**) and Rgs9 (**Krispel et al., 2006**) genes, and the two arrestins *Arrb1* and *Arrb3*, as the counterparts for the above-mentioned zebrafish genes for our analysis.

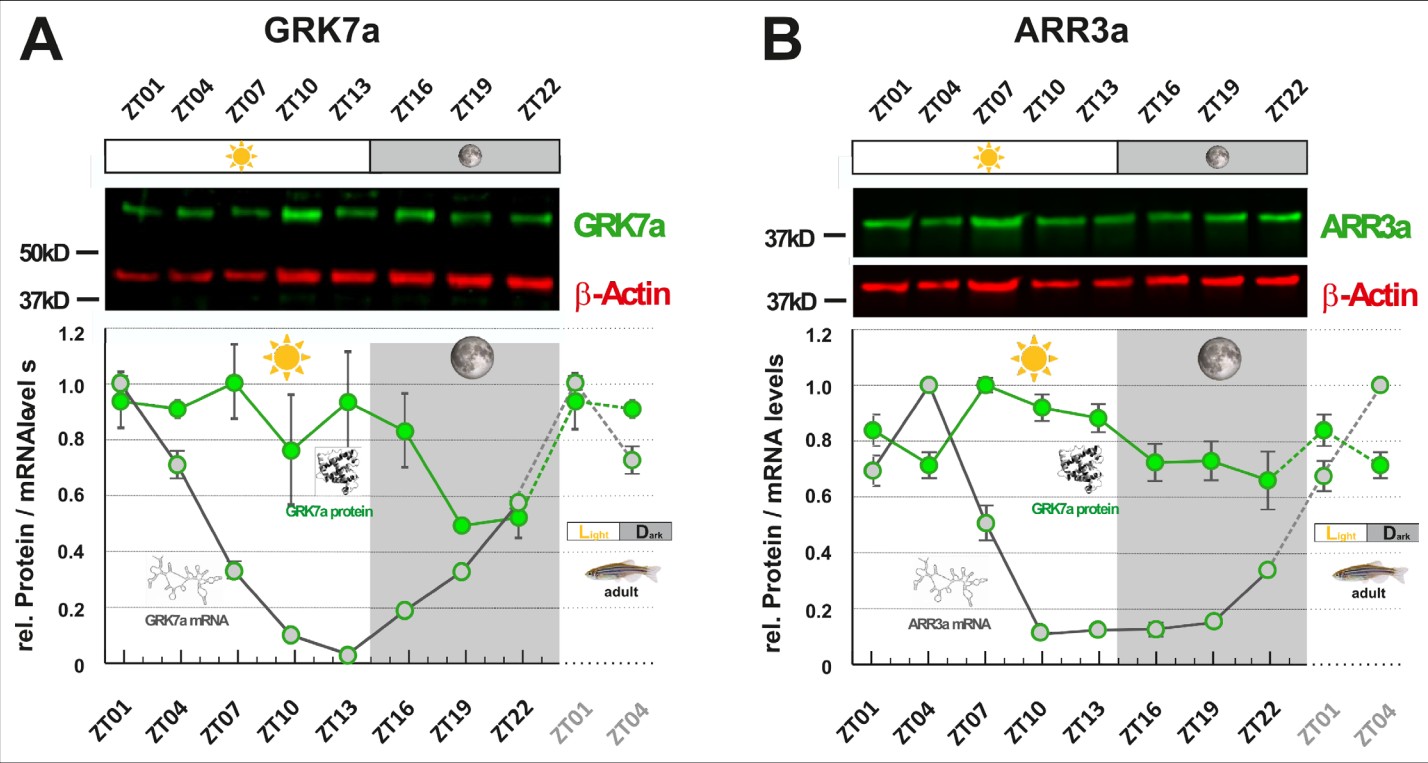

**Figure 3.** GRK7a and ARR3a protein levels show daily changes in adult zebrafish eyes. GRK7a (**A**) and ARR3a (**B**) protein levels were quantified using Western blot analysis. β-Actin was used as a loading control. While mRNA transcript levels (gray circles / RNA structure) were lowest in the evening (ZT10 and ZT13, respectively), lowest protein expression levels (green circles / protein structure) were tailing RNA expression levels by around 6 to 12 hours, reaching lowest levels in the middle of the night at around ZT19. The time point of collection is indicated along the x-axis with ZT01 being the time point one hour after the light was turned on. Dark periods are indicated by the moon symbol and highlighted in gray, whereas the periods under regular light conditions are indicated by the sun symbol and shown in white. Data represents the mean ± s.e.m. Statistical analysis was performed by "RAIN" as previously described (**Thaben and Westermark, 2014**). Statistics information and the numbers of independent repeats are provided in **Supplementary file 3**. Metadata can be downloaded from DRYAD.

The online version of this article includes the following figure supplement(s) for figure 3:

**Source data 1.** Protein levels of Grk7a and Arr3a in the eye of adult zebrafish were measured by infrared western blotting over a 24 hr period.

Expression of all five regulators fluctuated in a 24 hr period (**Figure 2**), being highest at the beginning of the dark period (ZT13) for the two arrestins (**Figure 2A&B**), or around midnight (ZT17) for *Grk1*, *Rgs1*, and *Recvrn* (**Figure 2C–E**). All of them displayed minimal transcript levels early during the day. This oscillation pattern shows a clear anti-phasic relationship with the cyclic fluctuation of the corresponding zebrafish transcripts. Curiously, the amplitude of gene fluctuation in adult zebrafish retina was generally larger than that in the mouse retina (**Figures 1 and 2**).

## Levels of key visual transduction regulator proteins fluctuate in the zebrafish retina

While mRNA half-life is typically in the range of minutes, protein turnover rates can range from minutes to days, explaining why fluctuation of mRNA levels is not always reflected in time-shifted oscillations at the protein level (**Cunningham and Gonzalez-Fernandez, 2000**; **Stenkamp et al., 2005**). However, as regulatory proteins often have turnover rates of only a few hours, we were examining whether RNA oscillations are mirrored by corresponding protein level fluctuations. In order to assess protein levels, we generated paralog-specific antibodies against GRK7a and ARR3a. Quantitative western blot analysis indicated periodic changes in protein levels for both proteins. Peak expression was shifted 6 - 12 hr between RNA and protein level (**Figure 3A&B**). ARR3a reached its highest and lowest levels at ZT7 and ZT22, respectively, whereas GRK7a maintained relatively high levels throughout the day, having

the lowest concentrations around midnight. Hence, mRNA circadian oscillations in the zebrafish retina are largely conserved at the protein level with a time shift.

## Larval cone response recovery is delayed in the evening

We next asked whether the observed protein and RNA level fluctuations have an impact on functional aspects of visual transduction. Photoresponses at larval zebrafish stages are dominated by cone photoreceptors (*Bilotta et al., 2001*). In the electroretinogram (ERG), the a-wave directly represents photoreceptor responses. Since in the zebrafish ERG, it is largely masked by the larger b-wave, reflecting the depolarization of ON-bipolar cells, we used the b-wave amplitude as an indirect measure of the cone photoresponse (*Figure 4A1*). The protein products of the genes analyzed in our study are known to affect photoresponse recovery in zebrafish (*Renninger et al., 2011*; *Rinner et al., 2005*; *Zang et al., 2015*). Therefore, we assessed their function by using the ERG double-flash paradigm. In this experimental setup, the retina receives a conditioning flash, followed by a probing flash of the same light intensity (*Figure 4A1*). The b-wave amplitude ratio of probing to conditioning response in relation to the interstimulus interval is a normalized read-out for the visual transduction recovery time (*Figure 4A2*; full example in *Figure 4—figure supplement 1*). Photoreceptor recovery is complete when the two flashes evoke responses of equal amplitudes. ERG responses are predicted to be contributed by all cone subtypes, given the light source spectrum.

Response recovery was significantly delayed in the evening in comparison to the morning (*Figure 4A2*). However, as the ERG b-wave is only an indirect measure of the photoreceptor response, we also measured the photoreceptor-induced a-wave by blocking the masking ERG b-wave (*Figure 4B1*). This was achieved by administering a pharmacological cocktail containing the excitatory amino acid transporter inhibitor DL-*threo*-beta-benzyloxyaspartate (DL-TBOA) and metabotropic glutamate receptor inhibitor L-2-amino-4-phosphonobutyric acid (L-AP4) (*Wong et al., 2004*). Consistently, the double-flash paradigm demonstrated that the a-wave response recovery in the evening was delayed (*Figure 4B2*). According to the light spectrum (*Figure 4—figure supplement 2*), the a-wave was contributed by all cone subtypes.

In order to prove that increased response recovery times measured in the evening are a bonafide circadian event, we repeated the above experiments on larvae that were kept in constant darkness. At corresponding time points, the decrease in response recovery was comparable (*Figure 4C1&C2*), verifying that the observed changes are regulated by an intrinsic circadian clock.

As photoresponse recovery is affected by the circadian rhythm, we hypothesized that this should also be apparent in temporal aspects of vision. Therefore, we recorded ERG responses generated by the flickering stimuli with different stimulus frequencies (*Figure 5*, 5 Hz, 8 Hz, 10 Hz, 12 Hz, and 15 Hz). Fast Fourier transform (FFT) algorithm in MATLAB was used to extract the power at stimulus frequency. This power was then normalized against the power at 50 Hz (line noise), which is far from the stimulus frequencies. In line with our hypothesis, we found that the normalized power at each stimulus frequency was significantly weaker in the evening compared with the power in the morning. This clearly indicates that the cone visual temporal resolution is under circadian control. Note here, the flicker ERG was mainly contributed by double-cone responses because of the spectral content of the stimulus light (*Figure 4—figure supplement 2*).

## Manipulation of gene expression by light is mirrored by functional changes

Next we measured larvae reared in a reversed light cycle (DL) where the night turns into a day. Under this condition, gene expression levels stayed in the fish's time. ISH for the genes of interest (*Figure 6A*) reflected this, with a stronger staining intensity in LD fish at 9 o'clock in the morning compared to DL fish at the same time. Consequently, when both groups were recorded at 120 hr post fertilization, a prolonged response recovery time was obtained in the fish maintained in reversed light cycle, reflecting the situation in fish kept in the normal light and recorded in the evening (*Figure 6D*).

While the intrinsic circadian clock is maintained in the absence of light, continuous light exposure has been shown to disrupt this intrinsic rhythm (*Laranjeiro and Whitmore, 2014*). We therefore evaluated if the circadian regulation of mRNA expression persists in larvae kept under constant light (LL). Strikingly, the gene expression differences between morning and evening detected under LD

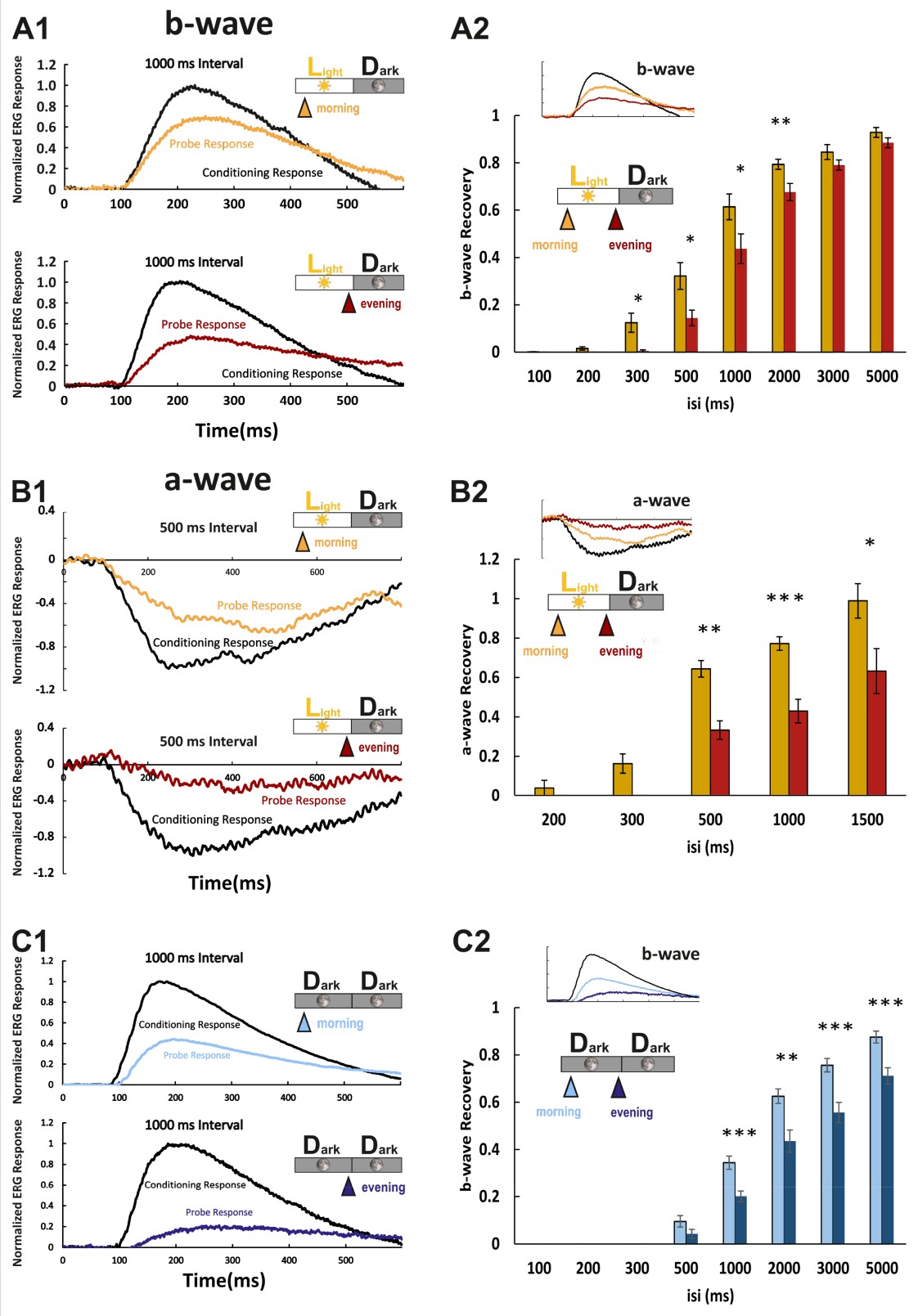

**Figure 4.** Larval cone photoresponse recovery is accelerated in the morning. (**A1**) Examples of normal light/dark (LD) larval electroretinogram (ERG) b-wave recordings. A conditioning flash (black line) was followed by a probing flash (yellow and red lines), which were separated by 1000 ms. While the yellow triangle and curve mark the probe response in the morning, the red triangle and curve represent the probe response recorded in the evening. Note that the probe response in the evening is clearly diminished. (**A2**) b-wave recovery as a function of the interstimulus interval (isi). At

*Figure 4 continued on next page*

*Figure 4 continued*

500 ms up to 3000 ms isi, b-wave recovery in the morning (yellow bars) is significantly enhanced when compared to corresponding recordings in the evening (red bars). Note that below 500 ms isi, no b-wave recovery can be observed and that at an interval of 5 s complete recovery can also be found in the evening. Data are presented as mean ± sem (n = 18 in the morning; n = 14 in the evening) of three independent experiments. t-tests and nonparametric tests were performed by GraphPad Prism version 8. p = 0.0149 at 300 ms isi; p = 0.0151 at 500 ms isi; p = 0.0405 at 1000 ms isi; p = 0.0069 at 2000 ms isi. *p<0.05; **p<0.01. (**B1**) Examples of LD larval ERG a-wave recordings under DL-*threo*-beta-benzyloxyaspartate (DL-TBOA) and L-2-amino-4-phosphonobutyric acid (L-AP4) inhibition. Under b-wave blocking conditions, a conditioning flash (black line) was followed by a probing flash (yellow and red lines), which were separated by 500 ms. The yellow triangle and curve mark the probe response in the morning, whereas the red triangle and curve represent the probe response recorded in the evening. Note that also the a-wave response recovery is significantly reduced in the evening. (**B2**) a-wave recovery as a function of isi. At 300 ms up to 1500 ms isi, a-wave recovery in the morning (yellow bars) is significantly enhanced when compared to corresponding recordings in the evening (red bars). Data are presented as mean ± sem (n = 11 in the morning; n = 5 in the evening) of three independent experiments. t-tests and nonparametric tests were performed by GraphPad Prism version 8. Plots with individual data points were provided in metadata from DRYAD. p = 0.0029 at 500 ms isi; p = 0.0003 at 1000 ms isi; p = 0.0375 at 1500 ms isi. *p<0.05; **p<0.01; ***p≤0.001. (**C1**) Examples of ERG b-wave recordings from a larva kept under constant darkness (DD). A conditioning flash (black line) was followed by a probing flash (light and dark blue lines), which were separated by 1000 ms. The light blue triangle and curve mark the probe response in the morning, whereas the dark blue triangle and curve represent the probe response recorded in the evening. (**C2**) b-wave recovery as a function of the isi is shown for larvae raised in continuous darkness (DD). Even under continuous darkness, visual function remains under circadian control as at 500 ms up to 3000 ms isi, and the b-wave recovery in the morning (light blue bars) is significantly enhanced when compared to corresponding recordings in the evening (dark blue bars). Data are presented as mean ± sem (n = 17 in the morning; n = 12 in the evening) of three independent experiments. t-tests and nonparametric tests were performed by GraphPad Prism version 8. p = 0.0007 at 1000 ms isi; p = 0.0016 at 2000 ms isi; p = 0.0004 at 3000 ms isi; p = 0.0006 at 5000 ms isi. *p<0.05; **p<0.01; ***p≤0.001. Metadata can be downloaded from DRYAD.

The online version of this article includes the following figure supplement(s) for figure 4:

**Source data 1.** Larval cone photoresponse recovery was measured by ERG in different conditions.

**Figure supplement 1.** An example of ERG recordings with the protocol used in *Figures 4 and 6*.

**Figure supplement 2.** Spectrum of ERG light.

conditions were completely lost in LL larvae (*Figure 6B&C*). This was also reflected on a functional level with no delay of photoresponse recovery in the evening, as measured by ERG.

Taken together, these results demonstrate that changes in the light cycle are reflected in changes of transcript levels of phototransduction regulators that subsequently lead to altered visual performance at different times during the day.

## Circadian clock-dependent expressions of key regulator genes tune the single-cone photoresponse kinetics

We applied a computational model of visual transduction to predict how the relative gene expression changes between morning and evening influence the single-cone photoresponse (*Invergo et al., 2013*; *Invergo et al., 2014*). The default model was set as morning value (ZT1). We then put the measured gene expression ratio data (*arr3a*, *grk7a*, *rcv2b* and *rgs9*) between ZT1 and ZT13 into the model for evening simulation. These four genes have been selected due to their pan-cone expression (*grk7a*, *rcv2b* and *rgs9*) and double-cone expression (*arr3a*), respectively. Running the model with the relative value of *arr3b* (blue and ultraviolet [UV] cones) produced comparable results to *arr3a* (data not shown). Detailed parameters are listed in *Supplementary file 4*. The computed morning and evening values were then compared.

As predicted by our experimental results, the decay of photoresponse to different light intensities in the model was largely prolonged in the evening (*Figure 7A–E*). The unsaturating response amplitude was slightly elevated in the evening, which may indicate the prolonged lifetime of the visual pigment (*Figure 7F*).

## Discussion

Circadian rhythms have been shown to regulate many biological aspects of vision. An early study demonstrated that zebrafish visual sensitivity is lower before light on and higher prior to light off (*Li and Dowling, 1998*). Later, another study linked the rhythmic expression of long-wavelength cone opsin to the core clock component CLOCK (*Li et al., 2008*). A particularly striking finding showed that

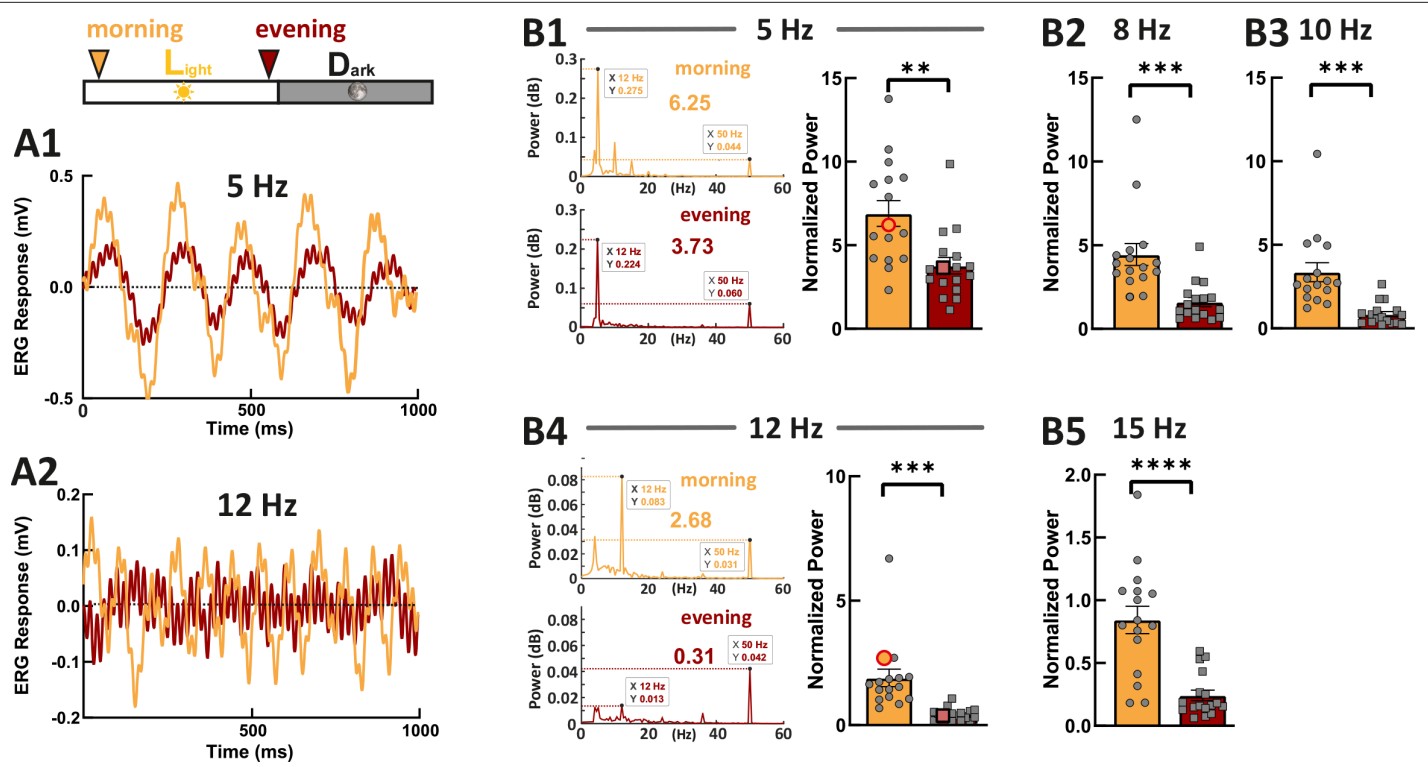

**Figure 5.** Zebrafish larvae show an increased temporal resolution in the morning. Examples show the flicker electroretinogram (ERG) responses to 5 Hz stimulus (**A1**) and to 12 Hz stimulus (**A2**). Example fast Fourier transform (FFT) power plots generated by MATLAB for responses (**A1**) and (**A2**) are shown in (**B1**) and (**B4**). These four example power plot results are highlighted in the corresponding summarized normalized power results in (**B1**) and (**B2**). The power of given frequency was normalized against the power at 50 Hz (line noise). The rest of the summarized plots of normalized power are shown in **B2**, **B3**, and **B5**. t-tests and nonparametric tests were performed by GraphPad Prism version 8. p = 0.0016 at 5 Hz (**B1**); p = 0.0005 at 8 Hz (**B2**); p = 0.0001 at 10 Hz (**B3**); p = 0.0001 at 12 Hz (**B4**); p<0.0001 at 15 Hz (**B5**). **p<0.01; ***p≤0.001; ****p≤0.0001. Metadata can be downloaded from DRYAD.

The online version of this article includes the following figure supplement(s) for figure 5:

**Source data 1.** Flicker ERG responses were measured.

synaptic ribbons of larval zebrafish photoreceptors disassemble at night. This peculiar phenomenon may save energy in fast-growing larvae (*Emran et al., 2010*). Our study now demonstrates that regulators of photoresponse decay are not only influenced by the circadian clock but in addition have a clear effect on the varying visual performances throughout a 24 hr cycle. Moreover, kinetics of cone visual transduction quenching is under the control of the circadian clock, which allows the fish to see with better temporal resolution in the morning than in the evening.

It is commonly assumed that circadian gene regulation helps the organism to optimally adapt to its preferential lifestyle and/or environment. Therefore, one would expect that the circadian systems of diurnal and nocturnal animals adapt differently. Our study indeed demonstrates that orthologous zebrafish and mouse genes involved in regulating cone visual transduction decay display an anti-phasic circadian expression pattern, supporting the functional relevance of the oscillating gene expression. While the visual temporal resolution of diurnal species is reduced in the evening, the visual system of nocturnal species is tuned to be most effective during these hours. Zebrafish, therefore, is an interesting model to study the physiology of circadian rhythms of diurnal animals, such as humans.

We would like to point out several additional interesting observations. Although many ohnologs (paralogs generated in a whole-genome duplication event), such as *grk7a* and *grk7b*, share a similar circadian phase or oscillatory amplitude, others, such as *rcv1a* and *rcv1b*, show an almost anti-phasic relationship. This is remarkable, since these ohnologs have been generated by a teleost-specific whole-genome duplication event (*Glasauer and Neuhauss, 2014*), implying that initially all ohnologs should have been in synchronicity. Interestingly, these ohnologs also adapted different expression

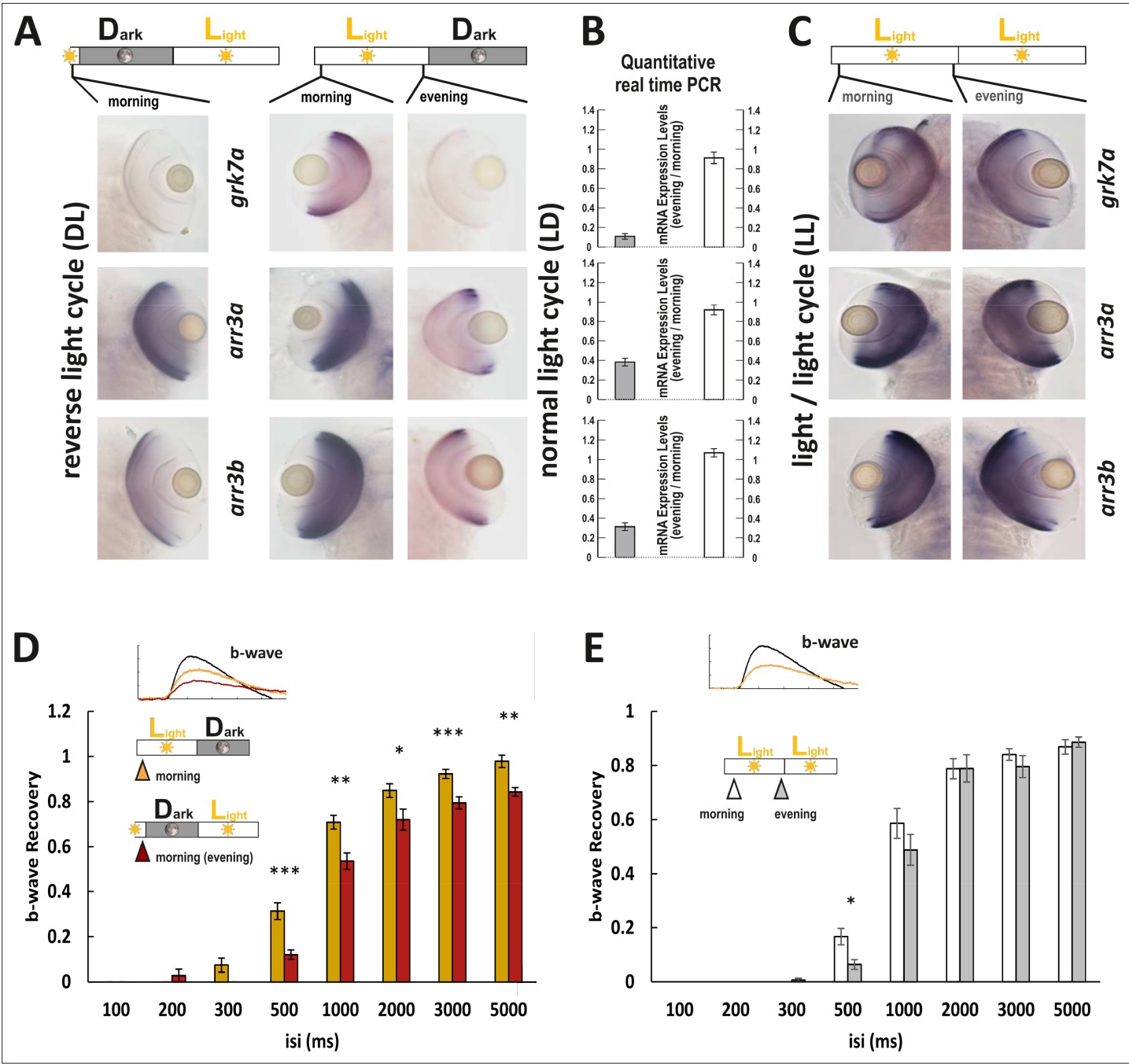

**Figure 6.** Light cycle alterations are reflected in adaptations of cone photoresponse recovery. (**A and C**) In situ hybridization images using *arr3a*, *arr3b*, and *grk7a* as probes. Tissues were collected from either reverse light cycle (DL) (**A**, left panel), normal light cycle (LD) (**A**, right panel) or light/light cycle (LL) (**C**) zebrafish larva (5 days post fertilization [dpf]) at the indicated time points. A reversal in the light cycle from LD to DL is reflected in the reversal of the in situ hybridization signal, with low expression levels observed at 9 o'clock (**A**). The ratio of gene expression levels between evening (ZT13) and morning (ZT1) for fish raised under a normal LD cycle or under LL is shown in (**B**). In contrast to the observed circadian regulation under LD conditions, under LL conditions, expression levels remain continuously elevated not displaying any circadian fluctuation (**B, C**). (**D**) A reversal of the light cycle is reflected in a corresponding reversal of b-wave recovery. The comparison of b-wave recovery of LD and DL larvae recorded at the same time in the morning clearly indicates that immediately before darkness, b-wave recovery rates are reduced. Data are presented as mean ± sem (n = 16 larvae raised in LD; n = 9 larvae raised in DL) of three independent experiments. t-tests and nonparametric tests were performed by GraphPad Prism version 8. Plots with individual data points were provided in metadata from DRYAD. p = 0.001 at 500 ms interstimulus interval (isi); p = 0.0019 at 1000 ms isi; p = 0.0221 at 2000 ms isi; p = 0.0009 at 3000 ms isi; p = 0.0022 at 5000 ms isi. *p<0.05; **p<0.01; ***p≤0.001. (**E**) No changes in b-wave recovery between morning and evening can be observed under constant light conditions (LL). Data are presented as mean ± sem (n = 15 in the morning; n = 12 in the evening) of three independent experiments. t-tests and nonparametric tests were performed by GraphPad Prism version 8. p = 0.0107 at 500 ms isi; *p<0.05.

*Figure 6 continued on next page*

*Figure 6 continued*

Metadata can be downloaded from DRYAD.

The online version of this article includes the following figure supplement(s) for figure 6:

**Source data 1.** Larval cone photoresponse recovery was measured by ERG in different conditions.

profiles, with *rcv1a* being expressed in rods and UV cones, while *rcv1b* being expressed in all cone types in the adult retina (UV, blue, red, and green) (***Zang et al., 2015***).

While the circadian rhythmicity of most genes persists throughout all developmental stages, some genes do show markedly different expression profiles between larval and adult stages. This may be

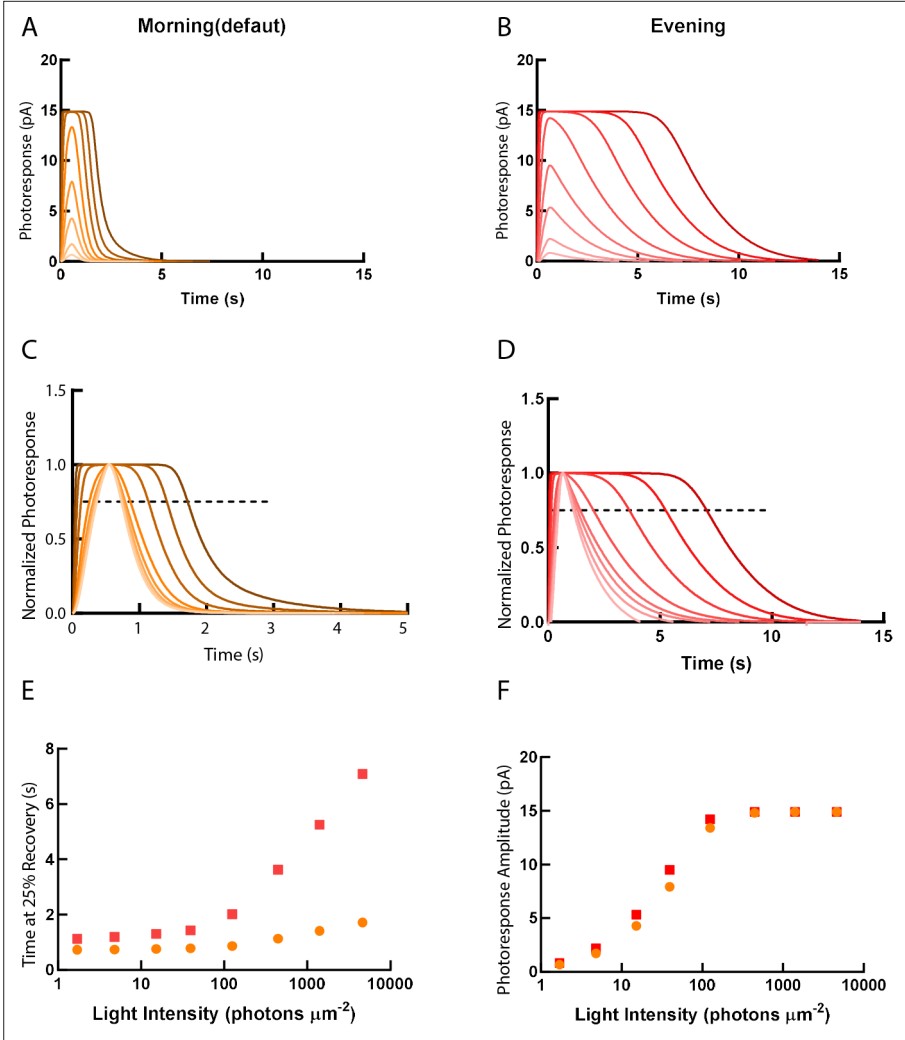

**Figure 7.** Simulations of single-cone photoresponse in the morning (default) and in the evening. Simulations of single cone photoresponse in the morning (default) (**A**) and in the evening (**B**). 500 ms flash stimuli were delivered at time = 0 s. The flash intensities are 1.7, 4.8, 15.2, 39.4, 125, 444, 1406 and 4630 photons μm$^{-2}$ (***Invergo et al., 2014***). (**C**) & (**D**) depict response curves normalized to the amplitudes at each light intensity. The dotted line represents 25% recovery of the photoresponse. Response duration for 25% recovery (**E**) and photoresponse amplitude (**F**) are plotted as a function of logarithmically increasing stimulus intensities.

The online version of this article includes the following figure supplement(s) for figure 7:

**Source data 1.** Single-cone photoresponse was predicted by a computational model.

**Figure supplement 1.** OKR measurements indicate increased contrast sensitivity at noon compared to evening but similar contrast sensitivity between morning and evening.

**Figure supplement 2.** Visual behavior shows differences between morning and evening.

related to the fact that the larval retina is functionally cone dominant, while the adult retina is a duplex retina with rod and cone contribution. In the case of *rcv2* ohnologs, *rcv2b* displays an in-phase cyclic expression pattern throughout all stages. Conversely, *rcv2a* did not show an overt cyclic expression pattern at larval stages, but being clearly under circadian control at adult stages (*Figure 1*). In contrast to the *rcv1* ohnologs, both *rcv2* genes are expressed in all cone subtypes, and depletion of either one acts to speed up the photoresponse termination (*Zang et al., 2015*). Other examples of ohnolog-specific cycling have been found for *arrs* and *rgs* genes (*Figure 1*, *Figure 1—figure supplement 4*). These observations strongly indicate that the transcription of clock-controlled genes (CCGs) is not uniformly regulated.

Interestingly, it has been previously demonstrated that the circadian clock seems to be desynchronized in larvae raised in darkness (*Dekens and Whitmore, 2008*; *Kaneko and Cahill, 2005*; *Kazimi and Cahill, 1999*; *Lahiri et al., 2014*). The circadian expression of some core clock genes and melatonin rhythms are lost when whole larvae were used as the experimental material in the absence of environmental entrainments. We did not observe this phenomenon in our study of visual transduction genes when only eye tissue was used, consistent with an inheritable maternal clock in the eye. We took care to avoid inadvertent environmental entrainment as described in detail in the 'Materials and methods' section. The different experimental results may come from the fact that different experiment materials were used. For example, all the analyzed genes in our study are also expressed in the photoreceptors of the pineal gland, but the transcript fluctuations may not necessarily be synchronized between the eye and pineal gland (eg, *rcv1a* in *Figure 1—figure supplement 1*). The use of whole larvae in our qRT-PCR study may have masked the cycling of retinal genes.

Furthermore, in many cases, the DD cycling is in phase with the fluctuations of the transcripts under the LD cycle. Endogenous circadian periods are around, but not exactly, 24 hr, and within 5 days in constant darkness, the peaks may shift relative to the LD cycle. In our experiments, the tissue was collected every 3 hr, and the shift within 5 days may be too small to be visible in the current experimental setting. The observation that the cycle of other genes (*arr3a*, *arr3b*, and *rcv1a*) in DD condition did diverge from LD condition indicates that these genes may be driven by different transcription factors. Furthermore, the DD condition led to the overall upregulation of some (eg, *rcv1a*, *rgs9a*) while caused downregulation of other (eg, *arr3a*, *arr3b*) genes as calculated from data in *Figure 1*. This strongly argues against a systematic error.

Among the studied genes in zebrafish, *grk7a* expression level increased by around 50 times in 1 day (*Figure 1A*), whereas Grk7a protein level increased by about two times in a 24 hr period (*Figure 3A*). *arr3a* transcript increased about 10 times (*Figure 1C*), while its protein level only grew less than 50 % throughout the day (*Figure 3A*). Therefore, these mRNA expression levels reflect proportionally to protein levels, indicative of a rather fast turnover rate for these proteins.

In the end, we asked whether the observed ERG adaptations between morning and evening directly influence visual behavior. Therefore, we measured the optokinetic response (OKR)(*Figure 7—figure supplement 1*) and the visual motor response (VMR)(*Figure 7—figure supplement 2*) . Both behavioral assays showed some changes between the different recording time points, but the direct contribution by visual transduction is hard to assign. Confounding factors, not related to vision, may for instance be circadian regulation of overall activity.

In conclusion, we have shown that key regulators of cone visual transduction at both the mRNA and protein level are under circadian control. Moreover, expression levels of these regulators in diurnal and nocturnal species are anti-phasic, suggesting that circadian changes influencing physiological and behavioral properties of vision are reflected in adaptation to different visual ecologies.

## Materials and methods
### Zebrafish care

Zebrafish (*Danio rerio*) were maintained at a standard 14 hr light:10 hr dark cycle (LD) with light on at 8 am and light off at 10 pm. Water temperatures were kept between 26 and 28 °C (*Amores et al., 1998*). Fish from the WIK wildtype strain were used in our study. Embryos were raised in E3 medium (5 mM NaCl, 0.17 mM KCl, 0.33 mM CaCl$_2$, and 0.33 mM MgSO$_4$) containing either 0.01 % methylene blue to suppress fungal growth and/or 0.2 mM 1-phenyl-2-thiourea (PTU; Sigma-Aldrich) to prevent pigment development. Embryos were collected directly after laying. LD condition embryos were then

**Table 1.** Sequences of primers used for qRT-PCR.

| | |
|---|---|
| ***rcv1a* S TGAGAACACGCCAGAAAAGC as CATTCAGGGTGTCATGGAGAAC** | |
| *rcv1b* s GCCTTCGCACTCTATGATGTG as CTCGTCGTCAGGAAGGTTTTTC | |
| *rcv2a* s CTTGGTCCTCTTTGGGAATCAG as AGTGGGCCTTCTCACTCTTC | |
| *rcv2b* s TGATGTGGACAAGAACGGTTAC as GGGAAGACTTGTCTGCTTGTC | |
| *arr3a* s GCCATCCCTTCACTTTCAATA as GCTTTTCCTTTGTCGTCTGG | |
| *arr3b* s ACTCCCCCTTGTTCTGATGTC as TTGCTCCTCACTGGCTGTAG | |
| *grk7a* s TGAACGTCTTGGCTGCAA as CCCAGGGTGGATCGATTAG | |
| *grk7b* s ACATTGAGGACCGCCTTG as CCCATGGAGGTGGAATGA | |
| *rgs9a* s CAACATTATAGGCCACGGATGAC as GATCCCTTCACACCAGTTGATG | |
| *ef1* s CTGGAGGCCAGCTCAAACAT as ATCAAGAAGAGTAGTACCGCTAGCATTAC (*Lin et al., 2009*) | |
| *actb2* s CCAGCTGTCTTCCCATCCA as TCACCACGTAGCTGTCTTTCTG (*Lin et al., 2009*) | |
| *rpl13* s TCTGGAGGACTGTAAGAGGTATGC as AGACGCACAATCTTGAGAGCAG (*Lin et al., 2009*) | |

transferred to the incubation room with normal light cycle (14:10). DD-conditioned embryos were placed in a black box before being transferred to the incubation room. Hence, all larvae (LD and DD) grew in the same environment with a stable temperature at 28 °C. LL-conditioned fish were raised under constant light. DL condition was light on at 8 pm and light off at 10 am.

Adult zebrafish were sacrificed using ice water following decapitation. All animal experiments were carried out in line with the ARVO Statement for the Use of Animals in Ophthalmic and Vision Research and were approved by the Veterinary Authorities of Kanton Zurich, Switzerland (TV4206).

## Zebrafish quantitative real-time PCR

Around thirty 5 dpf larvae or five eyeballs from adult zebrafish were collected per time point (ZT1, 4, 7, 10, 13, 16, 19 and 22) and the tissue stored in RNAlater (Sigma) at 4 °C. Dark adapted tissue was collected under dim red light. Only eyeballs were used for RNA extraction using the NucleoSpin RNA kit (Macherey-Nagel). Complementary DNA (cDNA) was produced using 110 ng total RNA as template for reverse transcription with SuperScript III (Invitrogen, Life Technologies; Zug, Switzerland). The samples collected from different time points were masked during RNA extraction and cDNA generation. qRT-PCR (Applied Biosystems Prism SDS 7900HT; Life Technologies) was performed using the MESA Green qPCR Mastermix Plus for SYBR Assay (Eurogentec, Seraing, Belgium) on a liquid handling robot platform (Tecan Genesis). Three technical replicates were conducted. Primers (Sigma-Aldrich) for qRT-PCR were intron-spanning to avoid amplification of non-digested genomic DNA fragments and were designed by online Universal ProbeLibrary Assay Design Center (Roche). Standard housekeeping genes (elongation factor 1, *ef1*; β-actin 2, *actb2* and ribosomal protein L 13, *rpl13*) were used as reference (*Tang et al., 2007*). Primer pairs used are listed in *Table 1*.Expression levels were normalized to 1. Statistical analysis was performed in R 4.1.0 with 'rain' package (*Thaben and Westermark, 2014*).

**Table 2.** Mouse primer sequences.

| | |
|---|---|
| ***Arrb1* S GCTCTGTGCGGTTACTGATCC as TGTCGGTGTTGTTGGTCACG** | |
| *Arrb3* s GCTAACCTGCCCTGTTCAGT as GCTAACCTGCCCTGTTCAGT | |
| *Grk1* s TGAAGGCGACTGGCAAGATG as AGGTCCGTCTTGGTCTCGAA | |
| *Rgs9* s TTCGCTCCCATTCGTGTTGT as ATGTCCTTCACCAGGGCTTC | |
| *Recvrn1* s AGTGGGCCTTCTCGCTCTA as ATCATCTGGGAGGAGTTTCACA | |
| *Actb* s CAACGGCTCCGGCATGTGC as CTCTTGCTCTGGGCCTCG | |

**Table 3.** Primer sequences for in situ probe preparation.

| | |
|---|---|
| *rcv1a* s GGACCAGAGTACAATTTAAG as GAAGCTCTAATCAGTCATAG (Zang et al., 2015) | |
| *rcv1b* s CAGACCAGCACCACATAC as TCTTGCACTTTCTGTGGTT (*Zang et al., 2015*) | |
| *rcv2a* s CAACATCTTTCTGAGCCC as ATAGCGTCTTCATTCTCC (*Zang et al., 2015*) | |
| *rcv2b* s CACTCAGACAGAAGTCAT as GTAGACCATCATCGCTTG (*Zang et al., 2015*) | |
| *grk7a* s GCATCTTCTAGTCTGATAGCAC as ACAGCTTCAATCATGTTAGTGA (*Rinner et al., 2005*) | |
| *grk7b* s CCCAGAGCGTCATATAGTG as AGTCACAGGAATAAGCTATGAA (*Rinner et al., 2005*) | |
| *rgs9a* s TTCCGGAATACAAAATGACAA as GCCTCGTGGGTCATTGAG | |
| *rgs9b* s GAAGCGAATATGACCATAAGG as ATCAGCCCTTCCTCGTTG | |
| *arr3a* s ATGGCTGACAAAGTTTACAAG as GCCCTGTGGAATCTGATATG (*Renninger et al., 2011*) | |
| *arr3b* s CATGACAAAGGTTTACAAGAAG as TGCTCCTCACTGGCTGTAG (*Renninger et al., 2011*) | |
| *arrSa* s CAATGAGTCCAAAAAATGTCG as TAACCGAGAAGTGCTCTTTC (*Renninger et al., 2011*) | |
| *arrSb* s ATGAGTCCCAAGCACATCATC as CAGCCAGCTCAAAACACG (*Renninger et al., 2011*) | |

## Mouse care and gene expression analysis

Mice were maintained at the Laboratory Animal Services Center (LASC) of the University of Zurich in a 12 hr light:12 hr dark cycle with lights on at 7 am. All animal experiments were performed according to the ARVO Statement for the Use of Animals in Ophthalmic and Vision Research and the regulations of Veterinary Authorities of Kanton Zurich, Switzerland.

Ten 12 -week-old wildtype mice (129S6; Taconic, Ejby, Denmark) were used in our experiments. Dark-phase mice were killed under red light and retinas were processed further under normal light conditions. Three mice at each time point (ZT1, 5, 9, 13, 17 and 21) were sacrificed and RNA was extracted (Macherey-Nagel, Oensingen, Switzerland) according to the manufacturer's instructions. cDNA synthesized using oligo-dT was done as previously described (*Storti et al., 2019*). The samples collected from different time points were masked during RNA extraction and cDNA generation. qRT-PCR was performed by ABI QuantStudio3 machine (Thermo Fisher Scientific) with the PowerUp Sybr Green master mix (Thermo Fisher Scientific). Two technical replicates were conducted. Primer pairs used are listed in *Table 2* for each gene of interest. Beta-actin (*Actb*) was used as a housekeeping gene to normalize gene expression with the comparative threshold cycle method (DDCt) using the Relative Quantification software (Thermo Fisher Scientific). The highest expression level was normalized to 1. Statistical analysis was performed in R 4.1.0 with 'rain' package (*Thaben and Westermark, 2014*).

## In situ hybridization

Primers used to generate in situ probes are listed in *Table 3*. Probes were digoxigenin-labeled using the DIG RNA Labeling Mix purchased from Roche.

For whole-mount ISH, embryos were treated with E3 containing 0.2 mM PTU (Sigma-Aldrich) to avoid pigmentation. 5 dpf larvae were fixed in 4 % paraformaldehyde (PFA; Sigma) in phosphate-buffered saline (PBS) overnight at 4 °C. Time points with maximal differences were chosen according to qRT-PCR results. Embryos were washed three times in PBS containing 1 % Tween (PBST), dehydrated step wise (25, 50, and 70 % methyl alcohol (MeOH) in PBST), and stored in 100 % MeOH at –20 °C. When comparing two groups of samples fixed at different time points, the tails of the group that may produce weaker staining were cut and mixed with the other group during staining.

For slide ISH, eyeballs were removed from adult zebrafish at different time points and fixed overnight at 4 °C using 4 % PFA. Detailed ISH processes have been previously described (*Haug et al., 2015*). When comparing two groups of samples fixed at different time points, both samples were placed on the same slide.

## Infrared western blotting

Five to six eyeballs from adult zebrafish were homogenized in ice-cold 150 ml RIPA buffer (150 mM NaCl, 1% Triton-X, 0.5 % sodiumdeoxycholate, 50 mM Tris (pH 8), 1 mM ethylenediaminetetraacetic acid [EDTA], 0.1 % sodium dodecyl sulfate [SDS]) containing cOmplete Protease Inhibitor Cocktail ([Roche]). After 2 hr of incubation on a 4 °C shaker, lysates were centrifuged for 30 min at 4 °C. During this procedure, all the samples were masked. Supernatants were stored at –80 °C. Nitrocellulose membranes with 0.45 μm pore size were used. Primary antibodies were diluted to the following concentrations: rabbit anti-Arr3a: 1:4000; rabbit anti-Grk7a: 1:3000; mouse anti-β-actin: 1:6000 (*Renninger et al., 2011*; *Rinner et al., 2005*). Anti-arr3a and anti-β-actin antibodies or anti-Grk7a and anti-β-actin antibodies were applied simultaneously. Secondary antibodies IRDye 800CW Goat anti-Rabbit IgG and IRDye 680RD Goat anti-Mouse IgG (LI-COR) were diluted in 1:20,000 ratio in blocking buffer (1 % bovine serum albumin [BSA] in PBST). Signal was detected by the Odyssey CLx Imaging System (LI-COR) and data were normalized to the internal loading control β-actin by IMAGEJ (*Schindelin et al., 2012*).

## Electroretinography

ERG was recorded as previously described (*Zang et al., 2015*). Light intensity (light source: Zeiss XBO 75 W) was measured using a spectrometer (Ocean Optics, USB2000b; software Spectra Suite, Ocean Optics) with a spectral range described previously (Supplemental Material 2A in *Zang et al., 2015*). Pairs of two light flashes with equal intensity and duration (500 ms) were applied (*Rinner et al., 2005*). Intervals between two flashes were either 100, 200, 300, 500, 1000, 2000, 3000, or 5000 ms. The interval between two pairs was 20 s. b-wave recovery is defined as the ratio of the second b-wave amplitude to the first one in the same pair.

To measure ERG a-wave, 5 dpf larval eyeballs were treated with 400 μM L-AP4 and 200 μM TBOA in Ringer's solution (111 mM NaCl, 2.5 mM KCl, 1 mM CaCl$_2$, 1.6 mM MgCl$_2$, 10 μm EDTA as a chelator for heavy metal ions, 10 mM glucose, and 3 mM 4- (2-hydroxyethyl) -1-piperazineethanesulfonic acid [HEPES] buffer, adjusted to pH 7.7–7.8 with NaOH). A HPX-2000 Xenon light source (Ocean Optics) was used and its light spectrum was measured by a spectrometer (Ocean Optics, USB2000b; software Spectra Suite, Ocean Optics; *Figure 4—figure supplement 2*). Electronic signals were amplified 1000 times by a pre-amplifier (P55 AC Preamplifier; Astro-Med. Inc, Grass Technology), digitized by DAQ Board (SCC-68; National Instruments), and recorded by a self-written Labview program (National Instruments). Intervals between two flashes were 300 ms, 500 ms, 1000 ms, and 1500 ms, respectively. a-wave recovery is defined as the ratio of the second a-wave amplitude to the first one in the same pair.

Flicker-fusion ERGs were measured with a white light emitting diode (LED) light source (Ocean Optics; LSM serie) controlled by LDC-1 controller (Ocean Optics). The spectrum of this light source was was measured by a spectrometer (Ocean Optics, USB2000b; software Spectra Suite, Ocean Optics; *Figure 4—figure supplement 2*). Except for the light source, flicker ERG was performed in the same setup as a-wave ERG. The flicker frequencies of 5 Hz, 8 Hz, 10 Hz, 12 HZ, and 15 Hz at 50 % duty cycle were used. Flicker-fusion ERG data were analyzed by MATLAB (R2020b).

## Phototransduction modeling

The computational model of vertebrate phototransduction was introduced and verified previously (*Invergo et al., 2014*; *Invergo et al., 2013*). We simulated the photoresponse to different light intensities of 1.7, 4.8, 15.2, 39.4, 125, 444, 1406, and 4630 photons μm$^{-2}$ with a flash duration of 500 ms. Default parameters in the model were kept for morning (ZT1) simulation. For evening (ZT13) simulation, the relative gene expression change between ZT1 and ZT13 of larvae LD conditions was applied. Parameters for each gene are listed in *Supplementary file 4*. The simulation was performed in COPASI (*Hoops et al., 2006*).

## Visual motor response

The VMR was measured using a Zebrabox (ViewPoint Life Science, Lyon, France). 5 dpf larvae were placed in a 96-well plate, subjected to dark adaptation for 10 min inside the Zebrabox, and the larval movement recorded with light off, on, and off for 5 min each. The distance that a single larva moved

was measured every 2 s. Baseline activity was calculated as the average movement 1 min before light on or off.

## Optokinetic response

The OKR was recorded as previously described (*Rinner et al., 2005*). Briefly, 5 dpf larvae were tested with sinusoidal gratings at different time points (ZT1, 4, 7, 10 and 13). To determine the contrast sensitivity, a spatial frequency of 20 cycles/360° and an angular velocity of 7.5 °/s were used with different contrast settings (5, 10, 20, 40, 70, and 100%). To explore the spatial sensitivity, an angular velocity of 7.5 °/s and 70 % of maximum contrast were applied with a varying spatial frequency (7, 14, 21, 28, 42, and 56 cycles/360°). Figures were prepared by SPSS (version 23.0; Armonk, NY: IBM Corp).

## Acknowledgements

We would like to thank Kara Kristiansen and Martin Walther for expert animal maintenance.

## Additional information

### Funding

| Funder | Grant reference number | Author |
|---|---|---|
| Schweizerischer Nationalfonds zur Förderung der Wissenschaftlichen Forschung | 310030_200376 | Marijana Samardzija |

The funders had no role in study design, data collection and interpretation, or the decision to submit the work for publication.

### Author contributions

Jingjing Zang, Conceptualization, Data curation, Formal analysis, Investigation, Methodology, Supervision, Validation, Writing - original draft; Matthias Gesemann, Methodology, Visualization, Writing - review and editing; Jennifer Keim, Investigation; Marijana Samardzija, Investigation, Validation, Writing - review and editing; Christian Grimm, Project administration, Supervision, Writing - review and editing; Stephan CF Neuhauss, Conceptualization, Funding acquisition, Project administration, Resources, Supervision, Writing - review and editing

### Author ORCIDs

Jingjing Zang (iD) http://orcid.org/0000-0002-2186-6001
Matthias Gesemann (iD) http://orcid.org/0000-0001-7635-1235
Christian Grimm (iD) http://orcid.org/0000-0001-9318-4352
Stephan CF Neuhauss (iD) http://orcid.org/0000-0002-9615-480X

### Ethics

All animal experiments were carried out in the line with the ARVO Statement for the Use of Animals in Ophthalmic and Vision Research and were approved by the Veterinary Authorities of Kanton Zurich, Switzerland (TV4206).

### Decision letter and Author response

Decision letter https://doi.org/10.7554/eLife.68903.sa1
Author response https://doi.org/10.7554/eLife.68903.sa2

## Additional files

### Supplementary files

- Supplementary file 1. Statistical information for *Figure 1*.
- Supplementary file 2. Statistical information for *Figure 2*.

- Supplementary file 3. Statistical information for *Figure 3*.
- Supplementary file 4. Parameters used in phototransduction model.
- Transparent reporting form

## Data availability

All data generated and analysed during this study are included in the manuscript and supporting files. The dataset has been uploaded to dryad at https://doi.org/10.5061/dryad.0cfxpnw26.

The following dataset was generated:

| Author(s) | Year | Dataset title | Dataset URL | Database and Identifier |
|---|---|---|---|---|
| Zang J, Gesemann M, Keim J, Samardzija M, Grimm C, Neuhauss SCF | 2021 | Circadian Regulation of Vertebrate Cone Photoreceptor Function | http://dx.doi.org/10.5061/dryad.0cfxpnw26 | Dryad Digital Repository, 10.5061/dryad.0cfxpnw26 |

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

# Appendix 1

## Appendix 1—key resources table

| Reagent type (species) or resource | Designation | Source or reference | Identifiers | Additional information |
|---|---|---|---|---|
| gene (*Danio rerio*) | arr3a | GenBank | BC076177 | |
| gene (*Danio rerio*) | arr3b | GenBank | BC059650 | |
| gene (*Danio rerio*) | grk7a | GenBank | BC163587 | |
| gene (*Danio rerio*) | grk7b | GenBank | AY900005 | |
| gene (*Danio rerio*) | rcv1a | GenBank | KT325590 | |
| gene (*Danio rerio*) | rcv1b | GenBank | KT325591 | |
| gene (*Danio rerio*) | rcv2a | GenBank | KT325592 | |
| gene (*Danio rerio*) | rcv2b | GenBank | KT325593 | |
| gene (*Danio rerio*) | rgs9a | GenBank | CABZ01019467 | |
| gene (*Danio rerio*) | actb2 | GenBank | AL929031 | |
| gene (*Danio rerio*) | ef1 | GenBank | L47669 | |
| gene (*Danio rerio*) | rpl13 | GenBank | AF385081 | |
| gene (*Mus musculus*) | Arrb1 | GenBank | AC102630 | |
| gene (*Mus musculus*) | Arrb3 | GenBank | AL671299 | |
| gene (*Mus musculus*) | Grk1 | GenBank | AC130818 | |
| gene (*Mus musculus*) | Rgs9 | GenBank | AK138159 | |
| gene (*Mus musculus*) | Recvrn | GenBank | CK617354 | |
| gene (*Mus musculus*) | Actb | GenBank | AC144818 | |
| sequence-based reagent | rcv1a s | This paper | qRT-PCR primers | TGAGAACACGCCAGAAAAGC |
| sequence-based reagent | rcv1a as | This paper | qRT-PCR primers | CATTCAGGGTGTCATG GAGAAC |
| sequence-based reagent | rcv1b s | This paper | qRT-PCR primers | GCCTTCGCACTCTATGATGTG |
| sequence-based reagent | rcv1b as | This paper | qRT-PCR primers | CTCGTCGTCAGGAAGG TTTTTC |
| sequence-based reagent | rcv2a s | This paper | qRT-PCR primers | CTTGGTCCTCTTTGGG AATCAG |
| sequence-based reagent | rcv2a as | This paper | qRT-PCR primers | AGTGGGCCTTCTCACTCTTC |
| sequence-based reagent | rcv2b s | This paper | qRT-PCR primers | TGATGTGGACAAGAAC GGTTAC |
| sequence-based reagent | rcv2b as | This paper | qRT-PCR primers | GGGAAGACTTGTCTGCTTGTC |
| sequence-based reagent | arr3a s | This paper | qRT-PCR primers | GCCATCCCTTCACTTTCAATA |
| sequence-based reagent | arr3a as | This paper | qRT-PCR primers | GCTTTTCCTTTGTCGTCTGG |
| sequence-based reagent | arr3b s | This paper | qRT-PCR primers | ACTCCCCCTTGTTCTGATGTC |
| sequence-based reagent | arr3b as | This paper | qRT-PCR primers | TTGCTCCTCACTGGCTGTAG |
| sequence-based reagent | grk7a s | This paper | qRT-PCR primers | TGAACGTCTTGGCTGCAA |

*Appendix 1 Continued on next page*

*Appendix 1 Continued*

| Reagent type (species) or resource | Designation | Source or reference | Identifiers | Additional information |
|---|---|---|---|---|
| sequence-based reagent | grk7a as | This paper | qRT-PCR primers | CCCAGGGTGGATCGATTAG |
| sequence-based reagent | grk7b s | This paper | qRT-PCR primers | ACATTGAGGACCGCCTTG |
| sequence-based reagent | rg9a as | This paper | qRT-PCR primers | CAACATTATAGGCCAC GGATGAC |
| sequence-based reagent | rgs9a as | This paper | qRT-PCR primers | GATCCCTTCACACCAG TTGATG |
| sequence-based reagent | ef1 s | *Lin et al., 2009* | qRT-PCR primers | CTGGAGGCCAGCTCAAACAT |
| sequence-based reagent | ef1 as | *Lin et al., 2009* | qRT-PCR primers | ATCAAGAAGAGTAGTACCGC TAGCATTAC |
| sequence-based reagent | actb2 s | *Lin et al., 2009* | qRT-PCR primers | CCAGCTGTCTTCCCATCCA |
| sequence-based reagent | actb2 as | *Lin et al., 2009* | qRT-PCR primers | TCACCACGTAGCTGTC TTTCTG |
| sequence-based reagent | rpl13 s | *Lin et al., 2009* | qRT-PCR primers | TCTGGAGGACTGTAAG AGGTATGC |
| sequence-based reagent | rpl13 as | *Lin et al., 2009* | qRT-PCR primers | AGACGCACAATCTTGA GAGCAG |
| sequence-based reagent | Arr1 s | This paper | qRT-PCR primers | GCTCTGTGCGGTTACTGATCC |
| sequence-based reagent | Arr1 as | This paper | qRT-PCR primers | TGTCGGTGTTGTTGGTCACG |
| sequence-based reagent | Arr3 s | This paper | qRT-PCR primers | GCTAACCTGCCCTGTTCAGT |
| sequence-based reagent | Arr3 as | This paper | qRT-PCR primers | GCTAACCTGCCCTGTTCAGT |
| sequence-based reagent | Grk1 s | This paper | qRT-PCR primers | TGAAGGCGACTGGCAAGATG |
| sequence-based reagent | Grk1 as | This paper | qRT-PCR primers | AGGTCCGTCTTGGTCTCGAA |
| sequence-based reagent | Rgs9 s | This paper | qRT-PCR primers | TTCGCTCCCATTCGTGTTGT |
| sequence-based reagent | Rgs9 as | This paper | qRT-PCR primers | ATGTCCTTCACCAGGGCTTC |
| sequence-based reagent | Rcv1 s | This paper | qRT-PCR primers | AGTGGGCCTTCTCGCTCTA |
| sequence-based reagent | Rcv1 as | This paper | qRT-PCR primers | ATCATCTGGGAGGAGT TTCACA |
| sequence-based reagent | Actb s | This paper | qRT-PCR primers | CAACGGCTCCGGCATGTGC |
| sequence-based reagent | Actb as | This paper | qRT-PCR primers | CTCTTGCTCTGGGCCTCG |
| chemical compound, drug | DIG RNA Labeling Mix | Roche | SKU11277073910 | |
| chemical compound, drug | 1-Phenyl-2-thiourea (PTU) | Sigma-Aldrich | CAS 103-85-5 | |
| chemical compound, drug | Paraformaldehyde (PFA) | Sigma-Aldrich | CAS 30525-89-4 | |

*Appendix 1 Continued on next page*

*Appendix 1 Continued*

| Reagent type (species) or resource | Designation | Source or reference | Identifiers | Additional information |
|---|---|---|---|---|
| chemical compound, drug | cOmplete, Mini, EDTA-free Protease Inhibitor Cocktail | Roche | SKU11836170001 | |
| chemical compound, drug | L-AP4 | Sigma-Aldrich | SKU A7929-.5MG | |
| chemical compound, drug | TBOA | Sigma-Aldrich | | |
| antibody | IRDye 680RD Goat (polyclonal) anti-Mouse IgG | LI-COR | P/N: 926–68070 | (1:1000) |
| antibody | IRDye 800CW Goat (polyclonal) anti-Rabbit IgG | LI-COR | P/N: 926–32211 | (1:1000) |
| antibody | Anti-arr3a (Rabbit polyclonal) | *Renninger et al., 2011* | | WB (1:250) |
| antibody | Anti-grk7a (Rabbit polyclonal) | *Rinner et al., 2005* | | WB (1:500) |
| antibody | Anti-β-Actin (Mouse monoclonal) | Sigma-Aldrich | A1978 | WB (1:1000) |
| software, algorithm | MATLAB | MATLAB(https://ch.mathworks.com/) | RRID:SCR_001622 | Version R2020b |
| software, algorithm | R | R (https://www.r-project.org/) | RRID:SCR_001905 | Version 4.1.0 |
| software, algorithm | COPASI | COPASI (http://copasi.org/) | RRID:SCR_014260 | |
| software, algorithm | Prism - GraphPad | GraphPad Prism (https://graphpad.com) | RRID:SCR_015807 | Version 8.0.0 |
| software, algorithm | Labview | National Instruments (https://www.ni.com/) | RRID:SCR_014325 | |
| software, algorithm | ImageJ | ImageJ (http://imagej.nih.gov/ij/) | RRID:SCR_003070 | |

