## [Decision Letter]

Thank you for submitting your article "Circadian Regulation of Vertebrate Cone Photoreceptor Function" for consideration by *eLife*. Your article has been reviewed by 3 peer reviewers, and the evaluation has been overseen by a Reviewing Editor and Didier Stainier as the Senior Editor. The following individuals involved in review of your submission have agreed to reveal their identity: Tom Baden (Reviewer #1); Pamela Menegazzi (Reviewer #3).

The reviewers have discussed their reviews with one another, and the Reviewing Editor has drafted this to help you prepare a revised submission. While we believe that your manuscript is in principle a very exciting advancement of the understanding of sensory perception, and thus likely well suited for publication in *eLife*, there were several critical comments and questions raised by the reviewers. Upon discussing the different reviewers' responses, we would ask you to address the following seven points below with particular importance. It is essential to properly address these points in order for us to consider your work for publication:

(1.) The finding that of the experiments examining retinal gene expression in larvae raised under constant darkness is at odds with several previous studies that show that robust circadian clock entrainment occurs only during the first days of zebrafish development and requires LD cycles for it. The finding of a maternal effect is (moderately phrased) "debated". The authors are aware of this, as they discuss this aspect in their own discussion:

line 7-9, page 22, "We did not observe this phenomenon in our study of visual transduction genes in the retina, suggesting the existence of an inheritable maternal clock in the eye (Delaunay et al., 2000).". At present their statement is a serious overinterpretation of their data, as it appears that they did not perform the controls to ensure constant environmental conditions. How was the DD collection really done? How about possible temperature changes the larvae might have been exposed to? Even small temperature changes can serve as entrainment cues.

Furthermore, it is actually rather suspicious that the DD cycling is pretty much in phase with the fluctuations of the transcripts under the LD cycle. Endogenous circadian periods are not exactly 24hrs and within 5days the peaks should have shifted relative to the LD cycle- which is not the case.

This point needs to be very carefully clarified.

(2.) Statistics and data presentation

The information on replicates is insufficient. Clearly define in all cases what "independent measurements" refer to and how many these are exactly in each case. Wherever n is below 10, please provide supplementary plots of the data with individual datapoints shown and refer to this in the respective figure legends.

Please also more carefully fill the transparent reporting form (e.g. information on group allocation (such as if masking was used?) is stated to be in the figure legends: Where is it?) Metadata can be downloaded from DRYAD, please add this as a statement to the Materials and methods section and below each Figure.

Importantly: In several cases these metadata are however NOT really self-explanatory and in part a little odd, e.g. Figure 5B- how do the metadata match with what is actually shown in the figure? (Also be aware that there are sometimes different labels in the Excel file itself versus its name. Or file names that could be interpreted as if the same fish and condition was tested twice, e.g. am: folder fish2: fish2_15_2 and fish2_15).

Statistics: Some stats are using tests that assume a normal distribution (e.g. t-test), but normality is not demonstrated (nor can it be in some cases as the n is too low). Please either verify such analyses with more robust statistical measures or explain why this would not be necessary.

(3.) Please improve the data presentation in Figure 5, which will likely benefit from a less binary analysis strategy. Why not compute FFTs of each response trace, and extract the power at stimulus frequency? This could be normalised if useful, for example against power at another frequency, or total power in the signal. This would allow plotting some consistent measure of response over circadian time for each stimulus frequency (yielding 5 curves). Presumably, there are also repeats in this data (n = 10 fish per condition? How many stimulus repeats?), which then would again allow quantifying variance as well.

(4.) The VMR experiments need to be much more carefully interpreted and discussed.

Firstly, it is not clear at all which of these aspects is due to visual sensation. In fact, zebrafish has 42 opsins, expressed at all kind of places. It has already been shown that eye-independent effects contribute significantly to the movement changes in response to sudden light changes (photokinesis).

Furthermore, the main difference (perhaps unsurprisingly) is that during the day larvae move more (i.e. higher baseline). When the light changes they presumably "twitch" which gives the peak, but since they are already moving, the relative peak is smaller than for fish at night, which move less as a baseline. Presumably, if fish use the same sort of motor programme in both conditions, they would, perhaps independent of baseline activity, move to a similar degree. This could be the simplest explanation of the fact that the two peaks are almost the same height. The subtle difference could then be explained by all kinds of other things ("warmed up" muscles in the light, possible circadian regulation of central circuits etc?). The OKR differences (Figure S7) are also quite subtle.

Taking these points together, it appears to be rather unclear how much of the differences can actually be linked to photoreceptor transduction differences in the eyes. The authors need to take all these points into consideration when interpreting their experiment and tone down their claims. The authors might consider to either remove Figure 7 entirely or to shift it into the Supplemental and instead consider a modelling approach. Numerous past phototransduction modelling studies provided amazingly functional frameworks that can be readily explored by "plugging in" some numbers of relative concentrations to see how this affects the photoreceptors. The authors have a beautiful collection of such numbers, for two ages in zebrafish and one in mice. What is the expected functional consequence for the photoresponse, based on all these cycling regulators? Something along these lines (not for a circadian context) was recently used e.g. for Figure 7 in Yoshimatsu et al., 2020 Neuron.

(5.) The presentation of the bioinformatics analysis for clock-regulated enhancer elements in the retinal gene promoters needs improvement. The data should be expanded significantly to show in detail, for each zebrafish gene, which enhancers have been identified and with which confidence (e.g. divergence from a consensus).

(6.) The graphical plots in Figures 1 and 2 are misleading. At first glance, they give the impression that samples have been collected over a 48 hours time course, which is incorrect. "Double plotting" of circadian data sets is often used in behavioural actograms, since it makes it easier to see trends in rhythmicity over longer time frames. However, in that situation the data is collected over multiple days and so rhythmicity is clearly demonstrated. In the case of gene expression reported here, a more honest way to present the data is, how it was done in Figure 3. One cycle has been sampled, NOT two, and so rhythmicity has not been directly visualised. However, given the amplitude of the changes in expression that have been observed during the 24 hours cycle, the sound statistical basis and the known regulatory input of the clock in the retina, it is acceptable to claim evidence for rhythmicity.

Furthermore, the authors should consider to use a more appropriate analysis to test for oscillations, for instance RAIN (Thaben and Westermark, J Biol Rhythms 2014). Given that RAIN does not make assumption on the waveform on the rhythms, it allows for testing of oscillation even under a light regimes that deviates from the standard LD12:12, as in this case. It should be possible to use the independent measurements as repetitions of the time series.

(7.) In 5 dpf larval zebrafish rods are non-functional. This should be mentioned somewhere, especially since results are to be compared to adults, or to mice, both of which have functional rods.

---

## [Author Response]

(1.) The finding that of the experiments examining retinal gene expression in larvae raised under constant darkness is at odds with several previous studies that show that robust circadian clock entrainment occurs only during the first days of zebrafish development and requires LD cycles for it. The finding of a maternal effect is (moderately phrased) "debated". The authors are aware of this, as they discuss this aspect in their own discussion:line 7-9, page 22, "We did not observe this phenomenon in our study of visual transduction genes in the retina, suggesting the existence of an inheritable maternal clock in the eye (Delaunay et al., 2000).". At present their statement is a serious overinterpretation of their data, as it appears that they did not perform the controls to ensure constant environmental conditions. How was the DD collection really done? How about possible temperature changes the larvae might have been exposed to? Even small temperature changes can serve as entrainment cues.Furthermore, it is actually rather suspicious that the DD cycling is pretty much in phase with the fluctuations of the transcripts under the LD cycle. Endogenous circadian periods are not exactly 24hrs and within 5days the peaks should have shifted relative to the LD cycle- which is not the case.This point needs to be very carefully clarified.

This is indeed an important point to clarify. We have detailed the procedure of sample collection and light treatment in the Material and Methods section and have added a paragraph to the discussion. Briefly, we detailed that our experimental conditions showed no evidence of inadvertent entrainment signals. The 5 day survey window is probably not enough to significantly deviate from the external rhythm for some genes. Indeed we found some that did shift, one even reversed the cycle. Hence, a systematic error is unlikely since genes are affected differently. Additionally, overall gene expression levels are also differently affected. Some genes were upregulated, while others showed a downregulation under DD conditions.

One important difference between our study and most published is that we collected eyes, while most studies used whole larvae, which includes other cycling tissues. We argued in the discussion that taking the whole larva may mask cycling of genes that show different expression cycling in different tissues. We show the example of rcv1a in the supplement (Figure 1, supplement Figure 1) where the transcript clearly cycles in the eye but is steady in the pineal.

(2.) Statistics and data presentationThe information on replicates is insufficient. Clearly define in all cases what "independent measurements" refer to and how many these are exactly in each case. Wherever n is below 10, please provide supplementary plots of the data with individual datapoints shown and refer to this in the respective figure legends.Please also more carefully fill the transparent reporting form (e.g. information on group allocation (such as if masking was used?) is stated to be in the figure legends: Where is it?)Metadata can be downloaded from DRYAD, please add this as a statement to the Materials and methods section and below each Figure.Importantly: In several cases these metadata are however NOT really self-explanatory and in part a little odd, e.g. Figure 5B- how do the metadata match with what is actually shown in the figure? (Also be aware that there are sometimes different labels in the Excel file itself versus its name. Or file names that could be interpreted as if the same fish and condition was tested twice, e.g. am: folder fish2: fish2_15_2 and fish2_15).Statistics: Some stats are using tests that assume a normal distribution (e.g. t-test), but normality is not demonstrated (nor can it be in some cases as the n is too low). Please either verify such analyses with more robust statistical measures or explain why this would not be necessary.

This is again is an important point that we changed. The statistics of all figures have been completely redone according current point (Figures 4 to 6) and according to point 6 (applying to Figure 1 to 3). The statistic information is now provided in supplement files 1 to 3 (Figure 1 to 3) and the rest in the metadata set.

We would like to add that this has really improved the manuscript and forced us to finally get more familiar with R.

(3.) Please improve the data presentation in Figure 5, which will likely benefit from a less binary analysis strategy. Why not compute FFTs of each response trace, and extract the power at stimulus frequency? This could be normalised if useful, for example against power at another frequency, or total power in the signal. This would allow plotting some consistent measure of response over circadian time for each stimulus frequency (yielding 5 curves). Presumably, there are also repeats in this data (n = 10 fish per condition? How many stimulus repeats?), which then would again allow quantifying variance as well.

This part has been completely redone according to the valuable suggestion. Although the reviewers were nice enough not to ask for additional experiments, we modified the light source to make it more controllable and performed new flicker ERG experiments with it.

(4.) The VMR experiments need to be much more carefully interpreted and discussed.Firstly, it is not clear at all which of these aspects is due to visual sensation. In fact, zebrafish has 42 opsins, expressed at all kind of places. It has already been shown that eye-independent effects contribute significantly to the movement changes in response to sudden light changes (photokinesis).Furthermore, the main difference (perhaps unsurprisingly) is that during the day larvae move more (i.e. higher baseline). When the light changes they presumably "twitch" which gives the peak, but since they are already moving, the relative peak is smaller than for fish at night, which move less as a baseline. Presumably, if fish use the same sort of motor programme in both conditions, they would, perhaps independent of baseline activity, move to a similar degree. This could be the simplest explanation of the fact that the two peaks are almost the same height. The subtle difference could then be explained by all kinds of other things ("warmed up" muscles in the light, possible circadian regulation of central circuits etc?). The OKR differences (Figure S7) are also quite subtle.Taking these points together, it appears to be rather unclear how much of the differences can actually be linked to photoreceptor transduction differences in the eyes. The authors need to take all these points into consideration when interpreting their experiment and tone down their claims. The authors might consider to either remove Figure 7 entirely or to shift it into the Supplemental and instead consider a modelling approach. Numerous past phototransduction modelling studies provided amazingly functional frameworks that can be readily explored by "plugging in" some numbers of relative concentrations to see how this affects the photoreceptors. The authors have a beautiful collection of such numbers, for two ages in zebrafish and one in mice. What is the expected functional consequence for the photoresponse, based on all these cycling regulators? Something along these lines (not for a circadian context) was recently used e.g. for Figure 7 in Yoshimatsu et al., 2020 Neuron.

This point is very well taken. We got slightly carried away by our enthusiasm. We gladly follow the suggestion and placed the original Figure 7 into the supplement. We ran the model as suggested and are happy to report that the recovery of phototransduction is indeed correlated to the cyclic gene expression as expected.

(5.) The presentation of the bioinformatics analysis for clock-regulated enhancer elements in the retinal gene promoters needs improvement. The data should be expanded significantly to show in detail, for each zebrafish gene, which enhancers have been identified and with which confidence (e.g. divergence from a consensus).

The detailed enhancer element analysis proved to be very extensive, mainly since only few gene regulatory genes in other vertebrate organisms are well annotated. We feel that the analysis is outside of the scope of this article and decided to remove the data.

(6.) The graphical plots in Figures 1 and 2 are misleading. At first glance, they give the impression that samples have been collected over a 48 hours time course, which is incorrect. "Double plotting" of circadian data sets is often used in behavioural actograms, since it makes it easier to see trends in rhythmicity over longer time frames. However, in that situation the data is collected over multiple days and so rhythmicity is clearly demonstrated. In the case of gene expression reported here, a more honest way to present the data is, how it was done in Figure 3. One cycle has been sampled, NOT two, and so rhythmicity has not been directly visualised. However, given the amplitude of the changes in expression that have been observed during the 24 hours cycle, the sound statistical basis and the known regulatory input of the clock in the retina, it is acceptable to claim evidence for rhythmicity.Furthermore, the authors should consider to use a more appropriate analysis to test for oscillations, for instance RAIN (Thaben and Westermark, J Biol Rhythms 2014). Given that RAIN does not make assumption on the waveform on the rhythms, it allows for testing of oscillation even under a light regimes that deviates from the standard LD12:12, as in this case. It should be possible to use the independent measurements as repetitions of the time series.

We performed the statistics and adjusted the plots accordingly (see also response to point 2).

(7.) In 5 dpf larval zebrafish rods are non-functional. This should be mentioned somewhere, especially since results are to be compared to adults, or to mice, both of which have functional rods.

This is indeed an important point to make. We have now referenced the functional cone only zebrafish retina and specified in each experiment which cone subtype contributes. Additionally we provide the emission spectrum of the light sources (Figure 4-figure supplement 2).

In some we were rightfully challenged (particularly on the statistics) by the reviewers and their insightful suggestions, including the model, improved the manuscript a lot.